# Human Adipose Tissue-Derived Mesenchymal Stromal Cells Inhibit CD4+ T Cell Proliferation and Induce Regulatory T Cells as Well as CD127 Expression on CD4+CD25+ T Cells

**DOI:** 10.3390/cells10010058

**Published:** 2021-01-01

**Authors:** Agnese Fiori, Stefanie Uhlig, Harald Klüter, Karen Bieback

**Affiliations:** 1Institute of Transfusion Medicine and Immunology, Medical Faculty Mannheim, Heidelberg University, German Red Cross Blood Service Baden-Württemberg-Hessen, 68167 Mannheim, Germany; agnefiori@gmail.com (A.F.); stefanie.uhlig@medma.uni-heidelberg.de (S.U.); harald.klueter@medma.uni-heidelberg.de (H.K.); 2FlowCore, Mannheim Medical Faculty Mannheim, Heidelberg University, 68167 Mannheim, Germany; 3Mannheim Institute for Innate Immunoscience, Medical Faculty Mannheim, Heidelberg University, 68167 Mannheim, Germany

**Keywords:** mesenchymal stromal cells, CD4 T cells, immunomodulation, regulatory T cells, CD127

## Abstract

Mesenchymal stromal cells (MSC) exert their immunomodulatory potential on several cell types of the immune system, affecting and influencing the immune response. MSC efficiently inhibit T cell proliferation, reduce the secretion of pro-inflammatory cytokines, limit the differentiation of pro-inflammatory Th subtypes and promote the induction of regulatory T cells (Treg). In this study, we analyzed the immunomodulatory potential of human adipose tissue-derived MSC (ASC), on CD4+ T cells, addressing potential cell-contact dependency in relation to T cell receptor stimulation of whole human peripheral blood mononuclear cells (PBMC). ASC were cultured with not stimulated or anti-CD3/CD28-stimulated PBMC in direct and transwell cocultures; PBMC alone were used as controls. After 7 days, cocultures were harvested and we analyzed: (1) the inhibitory potential of ASC on CD4+ cell proliferation and (2) phenotypic changes in CD4+ cells in respect of Treg marker (CD25, CD127 and FoxP3) expression. We confirmed the inhibitory potential of ASC on CD4+ cell proliferation, which occurs upon PBMC stimulation and is mediated by indoleamine 2,3-dioxygenase. Importantly, ASC reduce both pro- and anti-inflammatory cytokine secretion, without indications on specific Th differentiation. We found that stimulation induces CD25 expression on CD4+ cells and that, despite inhibiting overall CD4+ cell proliferation, ASC can specifically induce the proliferation of CD4+CD25+ cells. We observed that ASC induce Treg (CD4+CD25+CD127−FoxP3+) only in not stimulated cocultures and that ASC increase the ratio of CD4+CD25+CD127+FoxP3− cells at the expense of CD4+CD25+CD127−FoxP3− cells. Our study provides new insights on the interplay between ASC and CD4+ T cells, proposing that ASC-dependent induction of Treg depends on PBMC activation which affects the balance between the different subpopulations of CD4+CD25+ cells expressing CD127 and/or FoxP3.

## 1. Introduction

Mesenchymal stromal cells (MSC) have gained broad interest in regenerative medicine for their ability to modulate tissue repair and inflammation by secreting trophic and immunomodulatory factors. In general, their immunomodulatory activities have been reported to favor the skewing of the immune response towards a pro-regenerative and anti-inflammatory milieu [1,2]. In fact, MSC can modulate both innate and adaptive immune responses acting on several cell types of immune cells. In vitro, one of the hallmarks of MSC immunomodulatory activity is the suppression of CD4+ (T helper) but also CD8+ (cytotoxic) T cells proliferation, induced by mitogenic stimuli such as phytohaemagglutinin (PHA), anti-CD3/CD28, or allo-antigens [3,4,5]. Molecular mechanisms regulating MSC/T cell interplay rely on both cell-to-cell contact and secretion of soluble factors [6]. For instance, it has been demonstrated that MSC inhibited allogeneic T-cell responses in mixed lymphocyte reactions (MLRs) via indoleamine 2,3-dioxygenase (IDO) and tryptophan depletion [7]. In contrast, the interaction with antigen-presenting cells (APC) was found to be dependent on cell contact [8].

Similarly, in direct and indirect cocultures of MSC and peripheral blood mononuclear cells (PBMC), interleukin-10 (IL-10) and transforming growth factor beta (TGF-β) were involved in MSC-mediated T cell apoptosis [3]. Notably, MSC priming via interferon gamma (IFN-γ), resulted in IDO up-regulation which potentiated the suppressive potential of MSC on stimulated PBMC, suggesting that MSC activity can be modulated by surrounding stimuli [9,10].

Another important immunoregulatory function of MSC concerns their ability to affect the balance of the CD4+ T helper (Th) subsets both in vitro and in vivo [11]. For instance, in vitro cocultures of MSC and CD4+ cells showed that MSC could exert a potent inhibition of undifferentiated CD4+ T cells. However, once the differentiation to Th1 and Th17 was initiated, MSC suppressed only Th1 cells, while the pro-inflammatory Th17 were promoted [12]. In another study, MSC decreased IFN-γ secretion from Th1 cells while increasing IL-4 produced by Th2, suggesting a shift from a pro-inflammatory to an anti-inflammatory milieu [13]. Next to this, MSC modulate immune responses by inducing regulatory T cells (Treg) [14]. Treg are usually distinguished into natural or naïve Treg (nTreg), originating in the thymus, and induced Treg (iTreg), differentiated from peripheral CD4+ T cells [15]. Treg exert a fundamental role in the regulation of the immune response modulating the activation of Th subsets. They are involved in the prevention of autoimmune reactions and diseases, suppression of T cell activation and in the maintenance of self-tolerance [16]. Accordingly, a decreased frequency of Treg and/or reduced suppression functionality have been described in several disease contexts [17].

In general, three CD4+ T cell subsets can be classified based on the expression of the two cytokine receptors IL-2R (CD25) and IL-7R (CD127): (i) CD127+CD25low/− cells, which are IL-2-producing naive and central memory T cells; (ii) CD127−CD25− cells, considered as effector T cells expressing perforin and IFN-γ; and (iii) CD127lowCD25high cells, defined as FoxP3-expressing regulatory T cells (Treg) [18]. Cocultures of MSC and CD4+ cells promoted an induction of CD4+CD25+FoxP3+ Treg cells via prostaglandin E2 (PGE2) and TGF-β [19]. Treg formation was also induced by adipose tissue-derived MSC (ASC) in coculture with PBMC [2]. In another study, Melief and colleagues demonstrated that TGF-β, produced by MSC, was a key factor in mediating Treg induction [20]. In addition, IL-2, which is secreted mainly by activated effector T cells, has been demonstrated to be instrumental for Treg induction [21]. Many of these findings were recapitulated in several settings in vivo, highlighting the therapeutic effect of MSC via Treg induction [22,23].

Even though the immunomodulatory potential of MSC is to date commonly accepted, there are still various aspects, which are not fully elucidated or contradictory. This appears to be related to different experimental conditions and MSC sources [24]. For instance, in a study from Li and colleagues, MSC from Wharton’s jelly (WJ) were reported to exert a more potent inhibitory potential on T cell proliferation than ASC and even bone marrow (BM)-MSC [25]. In contrast, Najar et al. showed that ASC were the most immunosuppressive [26]. In addition, the use of whole PBMC or CD3+ or CD4+ fraction may affect results. Indeed, while CD4+CD25+ cells were increased in CD4+:BM-MSC cocultures [19], CD25 expression was reported to diminish in MSC:PBMC cocultures [27,28]. Lastly, even timing of cocultures, direct/indirect contact between the cells and cultures conditions, such as culture media and their additives, may represent critical points and affect results.

In this study, we analyzed the immunomodulatory potential of ASC in coculture with PBMC in vitro with a flow cytometry-based approach. Particularly, we developed direct and transwell cocultures with anti-CD3/anti-CD28 T cell receptor (TCR) stimulated or resting PBMC. Focusing on the CD4+ cell population, we first evaluated the inhibitory potential of ASC cocultures on CD4+ cell proliferation. Second, through cytokine analysis, we investigated T cell polarization and Treg induction. Finally, we evaluated the phenotypic expression of Treg markers such as CD25, CD127 and FoxP3 within CD4+ cells.

## 2. Materials and Methods

### 2.1. Generation of ASC and PBMC

ASC were isolated from multiple healthy donors undergoing liposuction after having obtained informed consent and characterized as described previously [29,30]. The study was approved by the Mannheim Ethics Commission II (vote numbers 2010-262 N-MA, 2009-210 N-MA, 49/05 and 48/05). After isolation, ASC were cultured at 200 cells/cm^2^ in DMEM (PAN Biotech; Aidenbach, Germany) supplemented with 10% of pooled human serum from AB donors (German Red Cross Blood Donor Service in Mannheim, Mannheim, Germany), 1% penicillin/streptomycin (PAN Biotech) and 2% L-glutamine (200 mM; PAN Biotech). Cells were maintained in incubators with controlled atmosphere (5% CO₂; 37 °C). Upon confluence, ASC were passaged with Trypsin/EDTA (PAN Biotech), counted and seeded according to the experiment. Cell morphology was constantly monitored (AxioVert100 Zeiss, Oberkochen, Germany). Prior to use, all ASC were assured to express the typical MSC immunophenotype and to differentiate into adipogenic and osteogenic lineages. All cells were cryopreserved in fetal bovine serum (FBS) with 10% of dimethyl sulfoxide (DMSO, Wak-chemie Medical GmbH, Steinbach, Germany) and were then thawed and cultivated for at least one passage before use. ASC from passage 2 to passage 3 were used in the experiments.

Human PBMC were isolated from buffy coat from healthy blood donors, provided by the German Red Cross Blood Donor Service in Mannheim. The Mannheim Ethics Commission II confirmed that no ethical approval was required for their use. PBMC were isolated freshly for each experiment with Ficoll-Paque™ (Ficoll-Paque™ Premium, GE Healthcare Bio-science AB, Uppsala, Sweden) density gradient isolation.

### 2.2. PBMC Staining before Coculture

To assess cell proliferation via serial dye dilution, 3 × 10^7^ freshly isolated PBMC were resuspended in PBS and stained with Violet Proliferation Dye 450 (VDP450, final concentration 1 μM, BD Biosciences, Heidelberg, Germany). After 15 min incubation at 37 °C, cells were washed in PBS, centrifuged and resuspended in RPMI 1640 supplemented with 10% FBS, 1% penicillin/streptomycin (PAN Biotech) and 2% L-glutamine (200 mM; PAN Biotech) (complete RPMI medium). According to the coculture conditions, PBMC were stimulated with anti-CD3/anti-CD28 loaded anti-biotin MACSiBead particle (T cell activation/expansion kit human; Miltenyi Biotec Gmbh, Bergisch-Gladbach, Germany) or not stimulated. A 1:2 bead-to-cell ratio was used, according to the manufacturer instructions. PBMC stimulated with beads or not were then directly used to setup cocultures.

### 2.3. PBMC:ASC Coculture

An amount of 2 × 10^5^ ASC were seeded in complete RPMI medium into a 6-well plate. Directly after isolation, 1 × 10^6^ VPD450 stained stimulated or not stimulated PBMC were added either (a) directly on top of the ASC monolayer (direct coculture) or (b) in a transwell insert (pore size 0.4 μm transparent ThinCerts-TC inserts; Greiner bio-one, Frickenhausen, Germany; transwell coculture). As control, VPD450-stained stimulated or not stimulated PBMC were seeded in identical conditions without ASC. A schematic representation of the experimental conditions is reported in Figure 1. Duplicates were run for each experiment. Controls and cocultures were treated with IL-2 (1 μg/mL, Promocell Gmbh, Heidelberg, Germany). After 7 days, cells were harvested and processed for analyses. Culture supernatants (conditioned media, CM) were collected and stored at −80 °C.

### 2.4. Flow Cytometry

After 7 days of culture, VPD450-stained stimulated/not stimulated PBMC were harvested from coculture and control conditions by resuspending them carefully. PBMC were then stained with antibodies reported in Table 1. All antibodies were properly titrated before use. PBMC were washed with PBS and resuspended in FACS Buffer. After 5 min incubation with 10 μL of FcR blocking reagent (human, Miltenyi Biotec Gmbh), anti-CD4, anti-CD25 and anti-CD127 (Table 1) antibodies for surface cell staining were added and incubated 20 min in dark at 4 °C. After washing, PBMC were resuspended in PBS, stained with Fixable Viability dye (eF780, 1:2000 final dilution; eBioscience, San Diego, CA, USA) and incubated for 30 min at 4 °C. For intracellular staining, cells were resuspended by vortexing in 1 mL of 1X Fix/Perm Buffer (BD Pharmingen™ Transcription Factor Buffer, BD Bioscience, Heidelberg, Germany) and incubated for 50 min at 4 °C. Then, cells were resuspended in 100 μL of 1X Perm/Wash buffer and stained with anti-FoxP3 antibody (Table 1). After 50 min of incubation at 4 °C, cells were washed twice with 1X Perm/Wash buffer and finally resuspended in FACS buffer. A volume of 100 μL of cell suspension together with 25 μL of precision count beads™ (Biolegend, San Diego, CA, USA) was analyzed immediately at BD FACS Canto II (BD Bioscience). Precision count beads were used to normalize FACS event acquisition and to calculate absolute cell numbers. The .fcs files were analyzed with FlowJo 10 software (FlowJo, LLC, Ashland, OR, USA).

#### Analysis of Division Index

To evaluate the extent of CD4+ cell proliferation, VPD450 stained PBMC were analyzed with the proliferation tool (FlowJo 7). As upon proliferation the VPD450 dye is distributed uniformly between daughter cells, it is possible to monitor cell division over time. The proliferation tool applies mathematical models to the proliferation data and develops statistics to describe it. Particularly, we used the division index (DI: average number of cell divisions that a cell in the original population has undergone, including those cells that never divided) to quantify CD4+ cell proliferation.

### 2.5. Cytokine Analysis

CM of cocultures and their respective PBMC controls were used to measure cytokine concentrations with either a multiplex bead array or ELISA.

#### 2.5.1. LEGENDplex™

Cell-free CM from 7 days stimulated/not stimulated direct and transwell cocultures and PBMC controls were measured using the LEGENDplex™ Human Th Cytokine Panel (13-Plex) (Cat. Number 740722; Biolegend) following the manufacturer’s instructions. Briefly, samples and standards were loaded in a 96-well V-bottom plate, followed by premixed beads addition. The plate was then covered and incubated for 2 h at RT under continuous shaking. After two washing steps with wash buffer, detection antibodies were added and incubated for 1 h at RT. LEGENDplex™ Streptavidin-PE was then added directly after the incubation and incubated for 30 min at RT. Further, another washing step was performed and samples were transferred to FACS tubes and analyzed at BD FACS Canto II (BD Bioscience). Data analysis was performed with the LEGENDplex™ Data Analysis Software (BioLegend). Data for IL-2, IL-4, IL-6 and IFN-γ are not reported since values were highly above (>6870.35 pg/mL) or below (<1.46 pg/mL) detection limit.

#### 2.5.2. ELISA

TGF-β was analyzed in CM of 7 days stimulated and not stimulated coculture and PBMC controls following the manufacturer’s instructions (Duo Set, Biotechne GmbH, R&D systems, Wiesbaden, Germany). ELISA plates were coated with the capture antibody diluted in PBS overnight at RT. The following day, the plate was blocked with reagent diluent and incubated for 1 h at RT. Then, standards and samples were added. The plate was incubated for 2 h at RT. Detection antibody was added for 2 h at RT. Finally, streptavidin-HRP was added for 20 min, followed by other 20 min of incubation with substrate solution (color reagent A and B, R&D systems). The plate was treated with stop solution (2N H_2_SO_4_). The optical density (OD) of each well was determined using a microplate reader (TECAN infinite M200PRO, Tecan Deutschland GmbH, Crailsheim, Germany) set to 450 nm. Wavelength correction was set to 570 nm. Standard curves were elaborated with GraphPad Prism 7 software (GraphPad Software, San Diego, CA, USA).

#### 2.5.3. Kynurenine Detection

To account for IDO enzymatic activity, kynurenine levels were measured in CM of 7 days stimulated/not stimulated cocultures and PBMC. An amount of 100 μL of probes and kynurenine standards (top standard 50 mM; Santa Cruz Biotechnology, Heidelberg, Germany) were distributed on a 96-well plate and mixed with 50 μL of 30% trichloroacetic acid (TCA; Carl Roth GmbH, Karlsruhe, Germany). The plate was incubated 30 min at 50 °C. After the incubation, the plate was centrifuged at 4004 g for 10 min. Without touching the pellets, 75 μL of supernatants were collected from each condition and transferred to a new 96-well plate. Then, 75 μL of 2% para-dimethylaminobenzaldehyde (Santa Cruz Biotechnology) were added and incubated for 15 min at RT. Finally, the OD of each well was determined using a microplate reader (TECAN infinite M200PRO) set to 492 nm. Standard curves were elaborated with GraphPad Prism 7 software. Kynurenine was not found in ASC cultured alone used as controls.

#### 2.5.4. IDO Staining

Indoleamine 2,3-dioxygenase (IDO) was measured in ASC cocultured with stimulated/not stimulated PBMC harvested by trypsinization after harvesting the PBMC at day 7. ASC were fixed for 30 min at RT (IC Fixation Buffer, eBioscience). After two washing steps, cells were stained in 1X permeabilization buffer with anti-IDO-PE (eBioscience,) and incubated for 30 min. Finally, cells were washed and analyzed immediately at BD FACS Canto II. ASC in monoculture were used as control. Data are presented as median fluorescence intensity (MFI) normalized on controls.

### 2.6. Statistics

All statistics were performed with GraphPad Prism v.8 software. Data are presented as mean ± standard deviation. Statistical comparison between the different experimental conditions are indicated by different symbols: * = stimulated vs. not stimulated; # = coculture vs. PBMC monoculture; § = direct vs. transwell.

*p* ≤ 0.05 is considered statistically significant (*/#/§ *p* ≤ 0.05, **/##/§§ *p* < 0.01, ***/###/§§§ *p* < 0.001 and ****/####/§§§§ *p* < 0.0001). *n* indicates number of biological replicates.

Box and whiskers: box extends from the 25th to 75th percentiles and whiskers go from min to max. Lines in the boxes are plotted at the median.

## 3. Results

### 3.1. ASC Abrogate CD4+ Cell Proliferation When Cocultured with Stimulated PBMC

To verify the immunosuppressive capacity of ASC, direct and transwell cocultures with anti-CD3/anti-CD28-stimulated and not stimulated PBMC were established and run for 7 days. As controls, stimulated and not stimulated PBMC monocultures (with and without transwell) were run. The inhibitory potential of ASC was corroborated by calculating the division index (DI) of CD4+ cells within the PBMC live population based on VPD450 dye dilution (Figure 2A). Upon PBMC simulation, ASC induced a 3.6- (direct) and a 2.9-fold (transwell) significant inhibition of the DI compared to stimulated PBMC monoculture (direct ### *p* < 0.001; transwell ## *p* < 0.01, Figure 2A’) obtaining DI values like those of not stimulated PBMC (stimulated PBMC vs. not stimulated PBMC: direct **** *p* < 0.001; transwell ** *p* < 0.01; Figure 2A’). No differences were found between direct and transwell within each condition. These data confirm that ASC exert an inhibitory function on CD4+ cell proliferation once PBMC are stimulated. Importantly, this inhibition is not dependent on cell contact.

The main mechanism involved in human ASC-mediated inhibition of T cell proliferation is the induction of IDO expression, which enzymatically degrades tryptophan, required for T cell proliferation, to kynurenine, a direct inhibitor of T cell proliferation [31]. To confirm this, IDO was measured in cocultured ASC and kynurenine concentrations in coculture CM. ASC isolated from stimulated cocultures expressed approximately 3- (direct) to 5.5-fold (transwell) more IDO than ASC from not stimulated cocultures (direct **** *p* < 0.0001; transwell ** *p* < 0.01, Figure 2B). Importantly, IDO was not detectable in ASC monocultures. Direct and transwell conditions did not differ. A similar trend was found for kynurenine concentrations in CM (Figure 2B’). Indeed, kynurenine was 1.2- and 1.5-fold more concentrated in stimulated cocultures than in not stimulated cocultures (direct ** *p* < 0.01; transwell *** *p* < 0.001, Figure 2B’). Importantly, in the not stimulated condition, kynurenine concentration was higher in direct than in transwell conditions (§§ *p* < 0.01, Figure 2B’) but this difference was not detected in the stimulated one. Kynurenine was not detected in CM of ASC and stimulated/not stimulated PBMC monocultures. In summary, these data show that, IDO levels are induced in ASC within stimulated cocultures in line with increased kynurenine levels.

All together, these data demonstrate the inhibitory potential of ASC on CD4+ cell proliferation within an anti-CD3/anti-CD28 stimulated PBMC population. High IDO induction in ASC and high kynurenine concentrations in the CM are indicative of tryptophan deprivation as the leading mechanism for inhibition of T cell proliferation. In addition, PBMC stimulation appears necessary to fully activate ASC inhibitory potential.

### 3.2. Cocultures Reduce Th Cytokine Secretion While Inducing TGF-β

MSC are known to affect Th subtypes, changing the balance of pro- and anti-inflammatory cytokines in vitro and in vivo [11]. To check potential ASC-mediated polarizing effects on T cell subsets, we analyzed the CM of cocultures. Upon anti-CD3/anti-CD28 stimulation the concentration of all cytokines was increased (from * *p* < 0.05 to *** *p* < 0.0001; Figure 3A–H). In contrast to the higher background level of proliferation in transwell cultures, cytokine levels were less induced than in direct cultures, especially for IL-5, IL-22 (both § *p* < 0.05), IL-17a and IL-17f (both §§ *p* < 0.01). In stimulated cocultures cytokine levels were significantly lower (from # *p* < 0.05 to ### *p* < 0.001) than in stimulated PBMC alone. Interestingly, ASC also reduced cytokine levels in not stimulated settings. Yet, it was not possible to identify any polarization toward anti- or pro-inflammatory cytokines and therefore to any specific Th subset. We conclude that upon PBMC stimulation, ASC reduce Th cytokine secretion reflecting the induced inhibition of T cell proliferation.

We further analyzed TGF-β levels within the CM (Figure 3I). In fact, TGF-β is often related to MSC/T cell interaction contributing to an anti-inflammatory switch of the Th subset via induction of Treg [32]. The TGF-β concentration was significantly higher in not stimulated cocultures than in stimulated ones (1.9-fold in direct, *** *p* < 0.001; 1.4-fold in transwell, not significant) and than in ASC alone (6.8-fold in direct, **** *p* < 0.0001; 3.3-fold in transwell cultures, * *p* < 0.05). TGF-β was not detectable in PBMC alone. These results clearly indicate that TGF-β secretion is particularly enhanced in not stimulated cocultures. Interestingly, in not stimulated cocultures, the content of TGF-β significantly differed between transwell and direct culture conditions (§§§ *p* < 0.001). This suggests that the direct contact with PBMC may enhance TGF-β secretion by ASC.

In summary, after 7 days of coculture, the analysis of cytokine profiles in CM was not indicative of any Th subset polarization in the PBMC population, neither in PBMC monoculture nor in cocultures. For all detected cytokines, we found that the TCR stimulation increased their secretion while cocultures had cytokine levels similar to that found in not stimulated PBMC monocultures. Importantly, TGF-β levels are predominantly increased in not stimulated cocultures.

### 3.3. CD4+CD25+ Cells Are Induced by Stimulation and Expanded by ASC

The cytokine profile analysis indicated that PBMC secrete high levels of cytokines, proving the TCR-stimulation induced activation. However, ASC reduced cytokine release in line with inhibiting T cell proliferation. Yet, surprisingly, when assessing CD25 expression, a marker of T cell activation [33], we observed a significant ASC-mediated increase in the fraction of CD4+CD25+ cells, gated within the live population of PBMC (Figure 4A,A’ and Appendix A). First as expected, stimulation increased the percentage of CD4+CD25+ cells 3-fold in direct PBMC compared to not stimulated PBMC monocultures (* *p* < 0.05; Figure 4A’). Then, however, upon stimulation ASC mediated a further 4.3- (direct) and 3.9-fold (transwell) increase in the CD4+CD25+ fraction (direct and transwell ** *p* < 0.01).

Stimulated cocultures had a 1.2- (direct) and 1.6-fold (transwell) higher percentage of CD4+CD25+ cells than PBMC alone (direct not significant, transwell # *p* < 0.05). These data document the stimulation-driven increase in CD25 on CD4+ cells (= activation) in PBMC monocultures. However, in contrast to our hypothesis, ASC even further increase the percentage of CD4+CD25+ despite inhibiting T cell proliferation and cytokine release.

We asked whether this increase resulted from the low level of proliferation observed in the cocultures (Figure 2A,A’). We investigated the CD4+CD25+ percentage of the proliferated (low VPD450 intensity) and the not proliferated (high VPD450 intensity) fraction within the live PBMC population (Figure 4B). Despite the few proliferating cells in cocultures upon PBMC stimulation, the percentage of proliferated CD4+CD25+ cells was 1.7- (direct) to 2.6-fold (transwell) higher than in PBMC alone (direct ### *p* < 0.001 and transwell #### *p* < 0.0001; Figure 4B’ and Appendix A) and higher than in not stimulated cocultures (12.6-fold in direct and 3.8-fold in transwell, both **** *p* < 0.0001). In the not stimulated cultures, the percentage of proliferated CD4+ CD25+ cells was at least 2-fold lower in cocultures (direct vs. transwell §§ *p* < 0.01) than in PBMC alone (direct ### *p* < 0.0001; transwell ## *p* < 0.01). Interestingly, stimulated and not stimulated PBMC monocultures yielded similar percentages of CD4+CD25+ cells in the proliferated fraction even though almost 95% of CD4+ cells proliferated upon stimulation and that only 20% to 30% of CD4+ cells did proliferate in the absence of stimulation. In addition, these data demonstrate that (1) ASC, despite their overall anti-proliferative action, promote some proliferation of CD4+CD25+ cells upon PBMC stimulation and (2) in absence of stimulation, ASC cause a significant reduction in the CD4+CD25+ fraction compared to PBMC alone.

In the not proliferated fraction (Figure 4B’’ and Appendix A), the percentage of CD4+CD25+ cells was approximately 4- (direct) and 3-fold (transwell) increased upon PBMC stimulation (direct ** *p* < 0.01; transwell * *p* < 0.05), indicative for their activation. In presence of ASC, this percentage appeared further increased, yet not significant compared to stimulated PBMC alone.

Overall, these data indicate that ASC, although preventing overall T cell proliferation and cytokine release upon stimulation, allow for activation of CD4+ cells and enable specific proliferation of this activated CD4+CD25+ fraction.

### 3.4. ASC Induce Treg and CD127 Upon Stimulation

So far, we have demonstrated that upon PBMC stimulation, ASC not only inhibited the proliferation and cytokine secretion of CD4+ cells, but also secreted TGF-β and promoted activated CD4+ (CD4+CD25+) cell proliferation. As TGF-β secretion and CD25 positivity are related to MSC-mediated induction of Treg [20], we calculated the percentage of CD4+CD25+CD127-FoxP3+ (Treg) cells, but within CD4+CD25+ cells, also the percentage of CD127-FoxP3- and CD127+FoxP3- subpopulations (Figure 5A, red, blue and green, respectively).

Upon PBMC stimulation, the Treg percentage was increased 1.7-fold in cocultures (red, direct ** *p* < 0.01; transwell * *p* < 0.05, Figure 5A’ and Appendix A) and 1.8-fold in PBMC monocultures (* *p* < 0.05) compared to their not stimulated counterparts. Yet, more striking was the 2.5- (direct) to 4.3-fold (transwell) increase in the percentage of CD127+FoxP3- cells in cocultures upon PBMC stimulation (green, direct # *p* < 0.05; transwell, ### *p* < 0.001). In addition, compared to not stimulated cocultures, this increase was significant (5.1-fold, direct and transwell *** *p* < 0.001). Finally, whereas anti-CD3/anti-CD28 PBMC stimulation led to a 4.3-fold increase in CD127-FoxP3- cells in PBMC monocultures (blue, **** *p* < 0.0001), this increase was 1.7-fold lower in presence of ASC (direct ### *p* < 0.001). Given that ASC appeared to reduce the percentage of CD4+CD25+CD127-FoxP3- cells in the not stimulated setting, the fold-change increase upon stimulation was similar in cocultures as in monocultures 4- (direct) to 5.4-fold (transwell) compared to not stimulated cocultures (direct *** *p* < 0.001; transwell **** *p* < 0.0001)).

All together, these data show that upon PBMC stimulation, cocultures induce a higher percentage of both Treg and CD127+FoxP3- cells within the CD4+CD25+ fraction. Importantly, the composition of the PBMC population is drastically changing in relation to ASC presence and PBMC stimulation, suggesting that these conditions may have a different impact on the induction of CD127 and FoxP3 expression.

### 3.5. ASC-Mediated Treg Induction Originate from a Variation in the Proportion of CD127/FoxP3 Expression within the CD4+CD25+ Subpopulation

To better characterize the dynamics of CD127 and FoxP3 expression in activated T cells, the same gating strategy reported in Figure 4B was applied now on the CD4+CD25+ subpopulation. We distinguished the proliferated CD4+CD25+ and not proliferated CD4+CD25+ and calculated the percentage of CD127±FoxP3± cells (Figure 6A,A’). Without stimulation (Figure 6A and Appendix A), we found that ASC led to a 3.6- (direct) and 1.4-fold (transwell) higher Treg percentage compared to PBMC alone (red, direct ### *p* < 0.001, transwell not significant). This percentage was also 2.2-fold higher than in stimulated cocultures (direct * *p* < 0.05). In addition, the fraction of proliferated CD127+FoxP3- was overall 2- to 2.7-fold higher in cocultures than in PBMC monocultures, yet here significantly induced upon stimulation (green, direct ## *p* < 0.01; transwell ### *p* < 0.001 and ** *p* < 0.01). In contrast, the percentage of proliferated CD127-FoxP3- cells was significantly reduced in cocultures: 1.7- (direct) to 1.8-fold (transwell) in the stimulated (blue, both ## *p* < 0.01) and 2.5-fold in the not stimulated condition (direct ### *p* < 0.001).

These data indicate that (1) ASC promote some proliferation of Treg in the not stimulated setting and (2) ASC, despite their inhibitory effect on stimulated PBMC, promote selective proliferation of the CD4+CD25+CD127+Foxp3- subpopulation.

In the subgroup of not proliferated CD4+CD25+ cells (Figure 6A’ and Appendix A), we found that upon PBMC stimulation, the Treg fraction was lowered, in both PBMC monocultures (red, 1.6- to 1.9-fold, direct * *p* < 0.05; transwell ** *p* < 0.01) and ASC cocultures (1.9- to 2.2-fold transwell and direct, respectively ** *p* < 0.01). Stimulated cocultures had a 1.3- to 1.1-fold lower percentage of not proliferated Treg than PBMC alone (not significant). Therefore, in contrast to the previous documented increase in Treg percentages in the proliferated fraction of not stimulated cocultures, no difference was observed here. Yet, a similar trend as in the proliferated fraction was seen regarding CD127+FoxP3- cells. ASC led to a 2.4- to 2.1-fold increase in not proliferated CD127+FoxP3- cells in direct and transwell cultures respectively (green, direct #### *p* < 0.0001; transwell ### *p* < 0.001) and a 2.6- to 2.8-fold in not stimulated direct and transwell cocultures (both ### *p* < 0.001). In line with that, the percentage of not proliferated CD127-FoxP3- cells was significantly lower in ASC cocultures than in PBMC alone (blue, all ## *p* < 0.01).

Compared to the previous analysis of CD4+CD25+ CD127±FoxP3± cells within the total PBMC live population (Figure 5A’), these data indicate that ASC and PBMC stimulation have clear distinct effects on the proliferation of subsets within the CD4+CD25+ population, affecting the balance between the different CD127±FoxP3± subgroups. Indeed, ASC promote Treg expansion in not stimulated conditions (Figure 6A) and, independent of stimulation, increase CD4+CD25+CD127+Foxp3- cells.

## 4. Discussion

In this study, we analyzed the immunomodulatory potential of ASC, evaluating their effects on CD4+ cells taking into consideration the activation state of whole PBMC and cell-contact. We established 7 day direct and transwell cocultures between ASC and whole stimulated and not stimulated PBMC and analyzed (1) the inhibitory potential of ASC on CD4+ cell proliferation and (2) phenotypic changes in CD4+ cells in respect of Treg marker (CD25, CD127 and FoxP3) expression. We found that ASC inhibited anti-CD3/anti-CD28-induced CD4+ cell proliferation, most probably via IDO–kynurenine activity, irrespective of direct and transwell cocultures. Cytokine analysis of CM revealed no specific polarization of Th subtypes not in stimulated PBMC nor in cocultures. While ASC reduced overall cytokine secretion upon PBMC stimulation, they also secreted TGFβ in particular in not stimulated cocultures, suggesting a possible Treg induction. To discriminate between ASC- and anti-CD3/anti-CD28 stimulation-mediated effects in the process of Treg generation, Treg markers were analyzed in respect of the total live population and of the proliferated/not proliferated cell fractions. We observed that (1) upon PBMC stimulation, CD25 expression is increased and the CD4+CD25+ population is further expanded by ASC, (2) ASC promote a selective increase in the proportion of CD4+CD25+CD127+FoxP3- cells independently of PBMC stimulation and (3) in not stimulated conditions, ASC increase the percentage of Treg.

Suppression of T-lymphocyte proliferation induced by cellular/non-specific stimuli was one of the first identified mechanisms of MSC-mediated immunomodulation [4]. Our data are in line with these previous observations and further confirm the inhibitory potential of MSC from adipose tissue [24]. In cocultures with resting PBMC, we did not observe an allostimulatory effect of ASC and the DI of CD4+ cell remained similar to that of resting PBMC monocultures. Similar findings were reported by Cuerquis et al. who demonstrated that resting PBMC were not affected by MSC in terms of suppression of proliferation [34], being in line with the hypothesis about the requirement of pro-inflammatory stimuli (or priming factors) to induce MSC-mediated immunosuppression [35,36]. However, these results are in contrast with the study from Crop et al. [37]. In fact, they observed that perirenal ASC, despite the lack of expression of HLA-class I and co-stimulatory molecules, did actually promote expansion and proliferation of T cells in resting PBMC, suggesting that ASC induced an allogenic response, like a mixed lymphocyte reaction (MLR) [37]. Maybe the use of ASC and IL-2 could explain these differences. In our cocultures, IL-2 addition was essential to prolong T cell expansion throughout the 7 days. Of note, preliminary experiments showed that IL-2 did not interfere with ASC-mediated suppression of T-cell proliferation even though some data suggest that addition of IL-2 to cocultures is able to abrogate MSC-mediated inhibitory effect [38].

MSC-mediated effects on T cells proliferation have been shown to be dependent on several mechanisms, which also involve the activity of IDO [39]. IDO is a catabolic enzyme which converts tryptophan to kynurenine [40]. In a previous study, we showed that increased IDO activity in ASC with subsequent tryptophan reduction and increasing kynurenine levels in their conditioned media was related to suppression of T cell proliferation in vitro [31]. In line with that, our data show that ASC from stimulated cocultures express higher levels of IDO than the ones from not stimulated cocultures.

MSC-immunomodulatory potential is not only related to inhibition of proliferation of activated cells. In fact, MSC have been reported to influence Th subset differentiation [11]. For instance, MSC were documented to promote the induction of anti-inflammatory and regulatory Th subsets in vitro [41,42,43] while inhibiting pro-inflammatory Th1 cells [1]. Measurement of Th cytokines in the conditioned media of cocultures allows the detection of a possible shift in the Th1/Th2 subpopulations [11,13]. Therefore, we analyzed coculture conditioned media looking for pro- (TNF-α, IL-17a, IL-17f and IL-22) and anti-inflammatory (IL-13, IL-5, IL-10, IL-9) cytokines. We found that independent of PBMC stimulation, both type of cytokines were generally reduced in cocultures compared to PBMC monocultures indicative of a general and indiscriminate inhibitory potential of ASC on cytokine secretion. This is clear in not stimulated cocultures, where the reduction in cytokines secretion was apparent. In line with that, in stimulated cocultures, the reduced cytokine levels reflect the inhibitory potential of ASC on CD4+ cell proliferation. Unfortunately, our data were not indicative of any pro- or anti-inflammatory switch and therefore any Th1/Th2 polarization. We speculate this may be related to the timing of cocultures. A polarizing effect of MSC on T cells has been reported already within 3 days after cocultures [12]. Among Th1/Th2 cytokines, we also measured TGF-β concentration since MSC-derived TGF-β was reported to be involved in Treg generation from CD4+ cells, contributing to the anti-inflammatory switch of T cells [19,44]. Similarly, we found that ASC secrete TGF-β and its secretion was significantly increased in cocultures compared to ASC monocultures and even more in not stimulated cocultures.

Next, supported by the finding of high TGF-β concentrations in cocultures, we focused on the characterization of Treg cells among the PBMC starting population analyzing the expression of some classical Treg markers such as CD25, CD127 and FoxP3 [45].

We found that upon PBMC stimulation the percentage of CD4+CD25+ cells was significantly increased in mono- and co-cultures despite the high inhibitory potential of ASC on CD4+ cell proliferation. The same was obtained when focusing on the percentage of CD4+CD25+ in the not proliferated fraction of PBMC, indicating that the majority of cells, which do not proliferate in cocultures, but highly proliferate in PBMC monocultures, express CD25. This led us to conclude that CD25 expression may be induced by the anti-CD3/anti-CD28 stimulation, being in line with some observations in the literature [46]. Interestingly, as the analysis of the proliferated fraction of PBMC revealed a significant high percentage of CD4+CD25+ in stimulated cocultures (although ASC inhibit proliferation to a large extent) but also a significant fraction in not stimulated cocultures, we hypothesized that ASC might promote the expansion of CD4+CD25+ cells. Indeed, English and colleagues showed similar outcomes reporting an increase in CD4+CD25+ cells in CD4+/BM-MSC cocultures [19]. Yet others reported reduced CD25 expression of stimulated PBMC or CD4+ cells cocultured with MSC [28,47]. We suggest therefore the following scenario where CD4+ cells normally express an extremely low level of CD25, which is increased upon TCR stimulation. The inhibitory potential of cocultures reduces CD4+ cell proliferation upon PBMC stimulation but at the same time ASC favor the expansion of CD4+CD25+ cells. We therefore propose CD25 as an activation marker and CD4+CD25+ cells more prone to proliferation when cocultured with ASC. Yet, Maccario et al. reported an MSC-induced increase in Treg using CD4+CD25+ as marker [48]. CD25 as Treg marker is still under debate, highlighting the need of more detailed studies on kinetic changes in CD25 expression with and without stimulation of PBMC/CD4+ T cells in presence of MSC.

To get closer to the Treg phenotype, we analyzed the expression of CD127 and FoxP3 in the CD4+CD25+ population. We found that the percentage of the Treg subpopulation was increased upon PBMC stimulation, in both mono- and co-cultures. However, a more detailed analysis of proliferated and non-proliferated CD4+CD25+ fractions revealed that (1) upon PBMC stimulation the percentage of Treg is lower in the not proliferated fraction and (2) without PBMC stimulation, ASC, especially in direct cocultures, yield a high percentage of Treg in the proliferated fraction. Overall, these data suggest that despite the low percentage of Treg detected, ASC induce a considerable fraction of Treg to proliferate in not stimulated cocultures but not once PBMC are stimulated. In contrast, Treg in the not proliferated fraction are diminished upon PBMC stimulation, even in presence of ASC suggesting that some “not induced” (probably “naïve”) Treg do not undergo proliferation. This ASC-mediated induction of Treg in not stimulated cocultures is in line with findings in the literature, despite different MSC sources, culture conditions and timing of the coculture [2,19,20]. This is also in line with the high TGF-β concentrations measured in the conditioned media of cocultures. Yet, our study proposes another point of view on ASC-mediated Treg induction: both ASC and anti-CD3/anti-CD28 stimulation affect the balance between the CD127±FoxP3± subpopulations, which inevitably affects the amount of Treg. Interestingly, even though CD4+ T cells highly proliferate upon PBMC stimulation, the relative composition of CD127 and FoxP3 cells/expression remains unchanged. However, within cocultures the composition changes significantly, reducing the proportion of CD127-FoxP3- cells (activated effector T cells) while increasing the CD127+FoxP3- (probably memory-like) T cells. Therefore, we postulate that ASC may induce the expression of CD127 on CD4+CD25+ cells independently of PBMC stimulation, contributing to changes in the proportions between CD127±FoxP3± subpopulations. The ASC-mediated increase in the percentage of the CD4+CD25+CD127+FoxP3- subtype may suggest the formation of memory T cells, marked by the co-expression of the two cytokine receptors IL-2R (CD25) and IL-7R (CD127). Importantly, this subset has been shown to secrete effector cytokines upon CD3 stimulation mainly of the Th2 type suggesting their specific function in controlling pro-inflammatory responses [49]. The modulation of effector/memory T cell subsets represents a novel point of interest in the study of ASC immunomodulation. A recent study showed that MSC-derived extracellular vesicles (EV) batch-dependently changed the composition of naïve T cells (T_N_; CD45RA+CCR7+), central memory T cells (T_CM_; CD45RA-CCR7+), effector T cells (T_E_, CD45RA+CCR7-) and effector memory T cells (T_EM_; CD45RA-CCR7-) after 4 h PMA/ionomycin stimulation of PBMC [50]. Yet here, one EV preparation reduced the frequency of CD4+ T_EM_ compared to stimulated PBMC without EV, putatively related to re-expression of CD45RA in memory T cells. Whether and how ASC modulate frequencies of naïve, effector and memory T cells is interesting to address in future studies.

We hypothesize that the ASC-dependent CD127 expression on CD25+ activated cells might be related to the secretion of IL-7 by ASC. The expression of CD127 is strictly dependent on IL-7 and IL-2 availability [51,52]. In the murine system, BM- MSC have been identified as major source of IL-7 which supported the proliferation and survival of colitogenic CD4 effector memory T (T_EM_) cells (CD4+CD45RBhigh T cells) [53]. These authors propose that the BM by secreting IL-7 provides a niche for CD4+ memory T cells. Yet, there are discrepant findings on IL-7-mediated effects on CD127±FoxP3± subpopulations. IL-7 has been reported to favor the formation of Treg [54] but also to limit their in vitro differentiation [55] and to inhibit the suppressive function of conventional Treg in favor of memory conventional T cells (CD4+CD25^low^CD45RO+FoxP3-) [56]. Importantly, our study points out a novel effect of ASC on the dynamics of the CD4+ T cell compartment.

## 5. Conclusions

Our study provides new insights on the interplay between ASC and CD4+ T cells. First, we confirmed the strong ASC inhibitory potential on CD4+ cell proliferation upon PBMC stimulation. This is mediated by IDO induction in ASC and concurrent kynurinine production. Second, we observed that CD25 expression is induced by stimulation. CD4+CD25+ cell proliferation occurs in cocultures even though ASC inhibit overall proliferation. Third, Treg induction and actual proliferation occurred only in not stimulated cocultures, supported by a high concentration of TGF-β. PBMC stimulation in contrast, appears to contribute to the reduction in the starting pool of Treg. Finally, ASC cocultures clearly affect the composition of the CD4+CD25+ subpopulations inducing the expression of CD127 and expanding the CD127+FoxP3- subpopulation. We propose IL-7 being secreted by ASC and contributing to the regulation of CD127 expression on CD4+CD25+ as main mechanism involved in ASC-mediated effects on dynamics of the CD4+ T cell compartment. This aspect may represent a new point of interest in the context of MSC-based therapeutic approaches in understanding how MSC exert their immunomodulatory potentials and affect the immune system in the context of the activation state of the surrounding immune cells.

## Figures and Tables

**Figure 1 cells-10-00058-f001:**
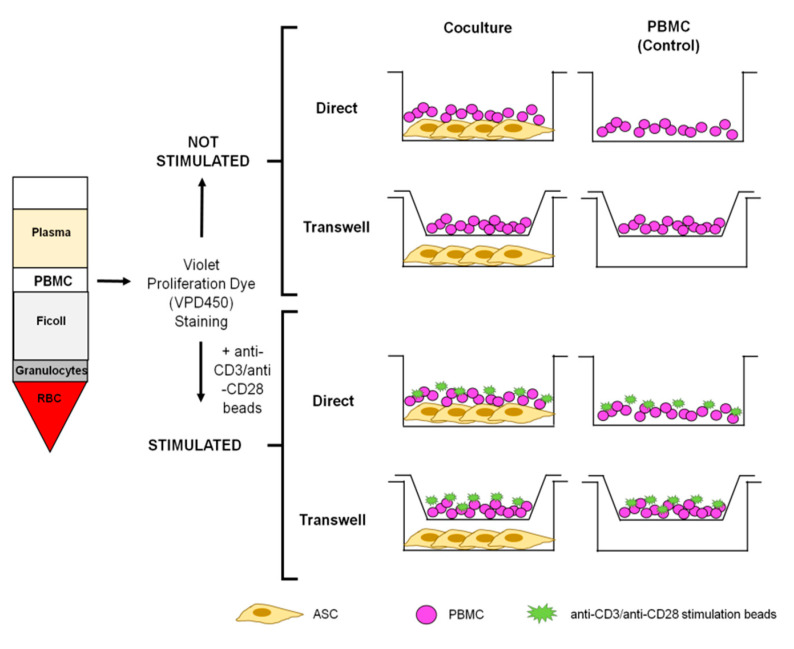
Schematic representation of the experimental conditions. Stimulated and not stimulated PBMC are seeded with or without ASC in direct and transwell cultures. PBMC monocultures represent the control conditions, respectively. ASC: adipose tissue-derived stromal cells; PBMC: peripheral blood mononuclear cells; RBC: red blood cells.

**Figure 2 cells-10-00058-f002:**
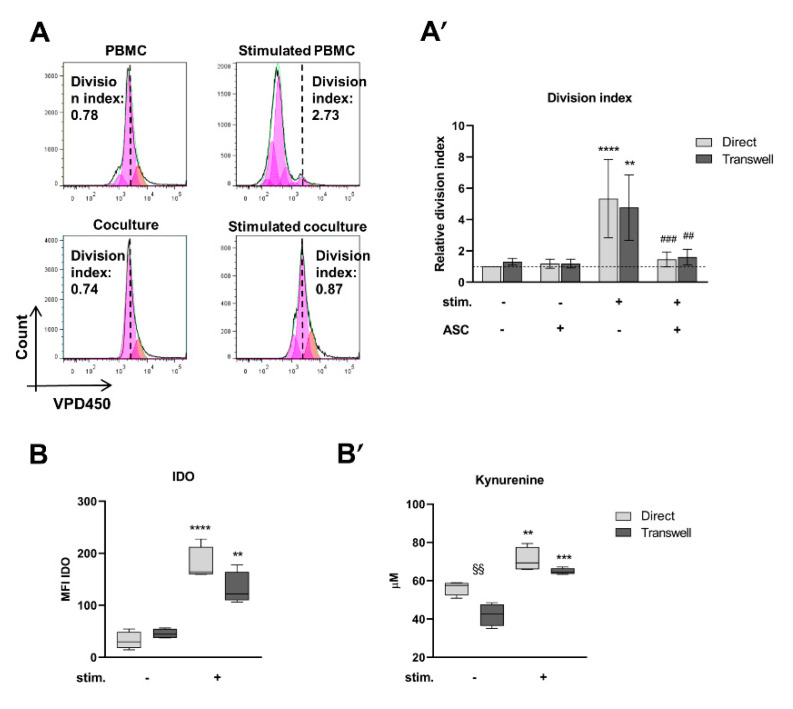
Adipose tissue-derived mesenchymal stromal cells (ASC) inhibit CD4+ cell proliferation. (**A**) Quantitative assessment of CD4+ cell proliferation using the Proliferation tool (FlowJo 7). First, the peak zero is set based on the VPD450 intensity in non-proliferated cells (orange peak). Then, the proliferation tool algorithm calculates the series of generations (pink peaks) and the division index (DI) to quantify proliferation. Dashed line: first generation peak. (**A′**) The DI of CD4+ within stimulated PBMC is significantly increased compared to controls (DI set to 1; direct: *p* < 0.0001, transwell: *p* < 0.01). ASC significantly reduce the DI in cocultures with stimulated PBMC (direct: *p* < 0.001, transwell: *p* < 0.01). *n* = 5, 2-way ANOVA multiple comparisons. (**B**) At day 7, indoleamine 2,3-dioxygenase (IDO) levels are significantly higher in ASC of stimulated cocultures compared to ASC of not stimulated cocultures (direct: *p* < 0.0001, transwell: *p* < 0.01). IDO was not detectable in ASC monoculture. *n* = 4, 2-way ANOVA multiple comparisons. (**B′**) At day 7, kynurenine levels in conditioned media (CM) of cocultures are higher in direct than in transwell not stimulated cocultures (*p* < 0.01) and significantly increased in the stimulated setting (direct: *p* < 0.01, transwell *p* < 0.001). Kynurenine was not detected in any monoculture CM. *n* = 4, 2-way ANOVA multiple comparisons.

**Figure 3 cells-10-00058-f003:**
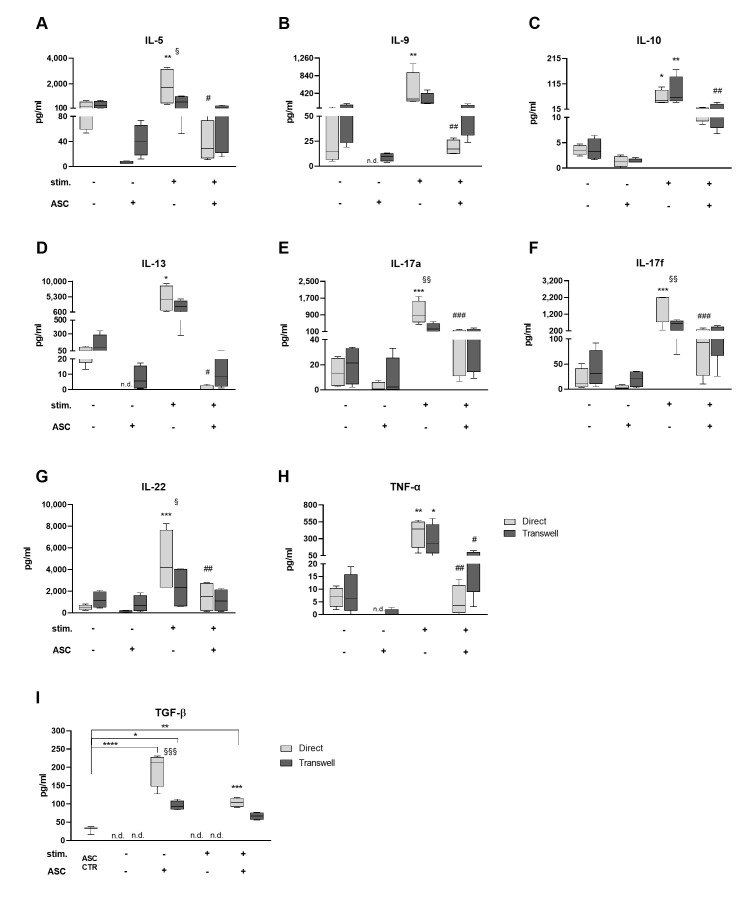
Cocultures reduce Th1/Th2 cytokine secretion and have increased levels of transforming growth factor beta (TGF-β). (**A**) Upon stimulation, interleukin (IL)-5 concentration is higher in direct PBMC (*p* < 0.01; direct vs. transwell *p* < 0.05) and direct cocultures (*p* < 0.05) compared to their controls. (**B**) Levels of IL-9 are more concentrated in stimulated direct PBMC than not stimulated PBMC (*p* < 0.01) and stimulated cocultures (*p* < 0.01). In not stimulated direct cocultures, IL-9 was not detectable (n.d.). (**C**) Upon stimulation, IL-10 is more concentrated in CM of PBMC than in CM of its controls (at least *p* < 0.05) and stimulated cocultures (transwell *p* < 0.01). (**D**) IL-13 concentration is higher in stimulated direct PBMC than not stimulated PBMC (*p* < 0.05) and direct stimulated cocultures (*p* < 0.05). IL-13 was not detectable in not stimulated direct cocultures. (**E**) IL-17a is significantly higher concentrated in CM of direct stimulated PBMC (direct vs. transwell *p* < 0.01) than not stimulated PBMC (*p* < 0.001) and stimulated cocultures (*p* < 0.001). (**F**) IL-17f is more concentrated in CM of stimulated direct PBMC (direct vs. transwell, *p* < 0.01) than not stimulated PBMC (*p* < 0.001) and stimulated cocultures (*p* < 0.001). (**G**) IL-22 concentration is higher in stimulated direct PBMC (direct vs. transwell, *p* < 0.05) than in CM of not stimulated PBMC (*p* < 0.001) and stimulated cocultures (*p* < 0.01). (**H**) TNF-α (tumor necrosis factor alpha) is significantly higher concentrated in CM of PBMC upon stimulation and presence of ASC (at least *p* < 0.05). TNF-α was not detectable in not stimulated direct cocultures. (**I**) Cocultures show significant higher levels of TGF-β compared to monocultures (at least *p* < 0.05). Levels are significantly higher in not stimulated condition (*p* < 0.001) and in direct coculture (*p* < 0.0001). n.d. = not detectable. *n* = 4, 2-way ANOVA multiple comparisons.

**Figure 4 cells-10-00058-f004:**
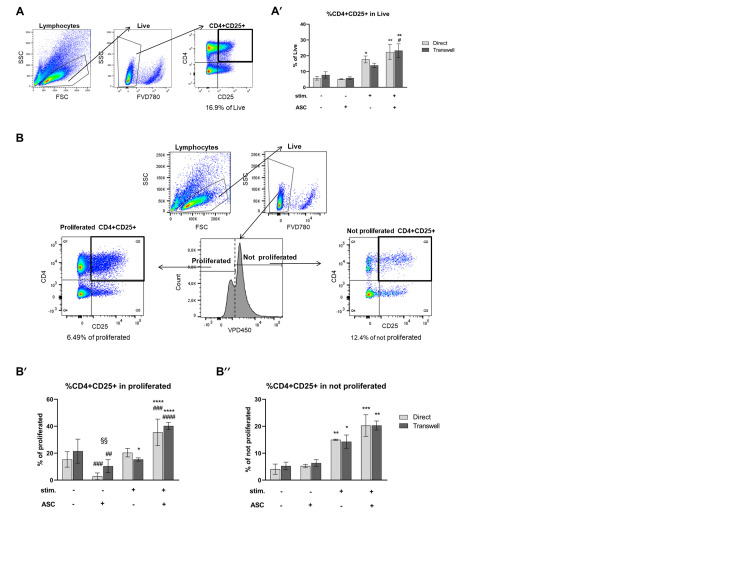
ASC increase CD25+ cells within the CD4+ compartment upon stimulation. (**A**) Representative gating strategy for CD4+CD25+ cells. Lymphocytes are gated based on sideward scatter (SSC) and forward scatter (FSC), then live cells are gated (FVD780 negative). Then, CD4+CD25+ percentage is calculated referring to the live population. One representative experiment is shown. (**A′**) Upon stimulation, CD25 expression is significantly increased (at least *p* < 0.05) and further increased by ASC cocultures (*p* < 0.05 in transwell). *n* = 3, 2-way ANOVA multiple comparison. (**B**) Representative gating strategy for proliferated and not proliferated CD4+CD25+ cells. Lymphocytes are gated based on SSC and FSC, and then live cells are gated (FVD780 negative). Proliferated and not proliferated cells are gated based on VPD450 intensity. Frequencies of CD4+CD25+ are then calculated within proliferated/not proliferated populations. One representative experiment shown. (**B′**) The combination of stimulation and cocultures induce significant CD4+CD25+ proliferation compared to the respective controls (at least *p* < 0.001). Contrarily, in not stimulated conditions, cocultures result in a significantly reduced proportion of proliferated CD4+CD25+ (at least *p* < 0.01 and direct vs. transwell *p* < 0.01). (**B′′**) Upon stimulation, the proportion of not proliferated CD4+CD25+ is significantly increased irrespective of ASC (at least *p* < 0.05). *n* = 3, 2-way ANOVA multiple comparison.

**Figure 5 cells-10-00058-f005:**
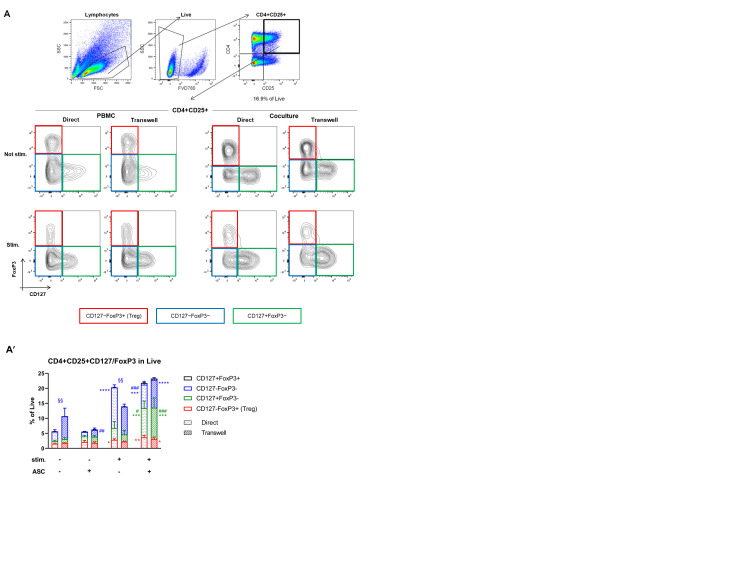
ASC induce regulatory T cells (Treg) and CD127+FoxP3- upon stimulation. (**A**) Representative gating strategy for Treg. Lymphocytes are gated based on SSC and FSC, then live cells are gated (FVD780 negative). Then, CD4/CD25 double-positive cells are gated and investigated for CD127 and FoxP3 expression. Red gate: CD127-FoxP3+ cells (Treg), blue gate CD127-FoxP3-, green gate CD127+FoxP3-. One representative experiment shown. (**A′**) Stimulation induces a slight increase in Treg (*p* < 0.05 in direct) which is further increased by ASC (at least *p <* 0.05). CD127+FoxP3- cells are increased in cocultures (not significant) but to a much higher extent in stimulated cocultures (at least *p* < 0.05). The percentage of CD127-FoxP3- cells is significantly reduced by ASC in both stimulated and not stimulated conditions (at least *p* < 0.01) with respect to their controls. Percentages are calculated referring to the starting live population. *n* = 3, 2-way ANOVA multiple comparison.

**Figure 6 cells-10-00058-f006:**
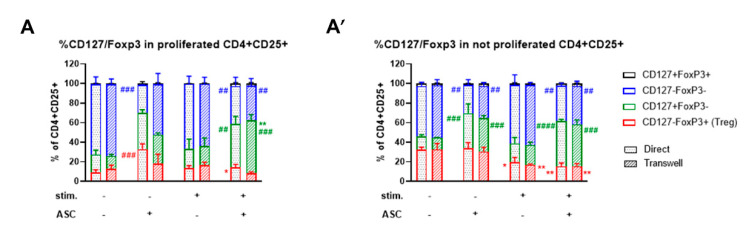
ASC significantly change the proportion of Treg and CD127+FoxP3- cells within the proliferated/not proliferated CD4+CD25+ subgroup. (**A**) The proportion of Treg within proliferated CD4+CD25+ cells is significantly increased in not stimulated cocultures (*p* < 0.001) but reduced in stimulated cocultures (*p* < 0.05). The percentage of CD127-FoxP3- cells within proliferated CD4+CD25+ cells is significantly lower in cocultures compared to their respective controls (at least *p* < 0.01) while the percentage of CD127+FoxP3- is higher (at least *p* < 0.01). (**A′**). Within the not proliferated CD4+CD25+ subgroup, the proportion of Treg is significantly reduced upon stimulation irrespective of cocultures (at least *p* < 0.05). The proportion of CD127+FoxP3- in the not proliferated cells is significantly increased in presence of ASC (at least *p* < 0.001). *n* = 3, 2-way ANOVA multiple comparison.

**Table 1 cells-10-00058-t001:** Flow cytometry-staining panel. VDP450 stained stimulated/not stimulated PBMC from controls and cocultures were harvested and stained following this panel.

Marker	Fluorochrome	Clone	Brand
Anti-CD4	FITC	RPAT4	BD Bioscience
Anti-CD25	APC	M-A251	BD Bioscience
Anti-CD127	PE-Cy7	REA614	Miltenyi Biotec
Anti-FoxP3	PE	259D/C7	BD Bioscience
Fixable Viability dye	eF780		eBioscience
Violet Proliferation dye	VPD450		eBioscience

## Data Availability

All data are included in the paper. There are no databases associated with this manuscript.

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
