# Peer review of "Human Adipose Tissue-Derived Mesenchymal Stromal Cells Inhibit CD4+ T Cell Proliferation and Induce Regulatory T Cells as Well as CD127 Expression on CD4+CD25+ T Cells"

_cells, 2021, doi:10.3390/cells10010058_

Round 1

Reviewer 1 Report

This work study the effects of ACS on CD4 T cells under resting and stimulated conditions.

Although the work is not completely novel, yet it provides a nice insight into the differential effect of ACS on resting vs. stimulated T cells. The in vitro work is well done. However, the interpretations are not clear enough and may miss some important observations.

Considering that:

In stimulated (experienced cells) CD127 is mostly a marker of memory CD4+ cells, which is expressed also at low level on Treg cells and may increase upon stimulation. Therefore, CD127-FoxP3- are effector T cells whereas CD127+FoxP3- are most likely memory-like T cells.

CD25, is a marker for Treg cells, but as part of the high affinity of IL-2R, it is also expressed on recently stimulated effector CD4 T cells. Therefore, CD4+CD25+ are stimulated effector cells, as well as Tregs.

Altogether, it seems that ACS induce differential immunomodulation; Whereas ACS support selective proliferation of Treg cells (FoxP3+) under resting condition, following stimulation they promote restricted proliferation of memory-like T cells (CD4+CD25+CD127+FoxP3+) and lowered the proliferation rate of Treg cells (maybe by combination of IDO-kynurenine activity, IL-7 and other factors). The monoculture, as expected, increases the percentage of stimulated effector cells CD127-FoxP3-. The effect on generation/maintenance of memory-like T cells do not distinguish between the different subtypes of CD4+ cells (Th1/Th2/Th17), which are skewed by alternative cues.

I think that the insights should be re-write

Reviewer 2 Report

The manuscript entitled “Human adipose tissue-derived mesenchymal stromal cells inhibit CD4+ T cell proliferation and induce regulatory T cells as well as CD127 expression on CD4+CD25+ T cells” analyzed the immunomodulatory potential of human adipose tissue-derived MSC (ASC) on CD4+ T cells; evaluated the phenotypic changes induced in CD4+ cells in respect of Treg marker (CD25, CD127 and FoxP3) expression; assessed IDO/Kynurenine involvement. This study, by Fiori and colleagues, proposed that ASC induction of Treg depends on the activation status of PBMC as well as on the balance between the different CD4+CD25+ subpopulations.

This article could be a new contribution in the knowledge on the immunomodulatory features induced by ASC in the modulation of both innate and adaptive immune responses; however this paper offers only few new information and few new point of view on the topic. Most of the contents in the manuscript are well known by the readers and previously summarized in other articles. Authors should make the article stronger and more appealing for publication by revising the discussion and making results section more fluent.

Below are reported my suggestions to the authors:

In general, I suggest the authors to re-edit the manuscript in order to make it fluent; the discussion should be better presented; the manuscript should be proofread.

ABSTRACT

  • The abstract must be revised balancing the background (too long) and implementing the results and purpose sections; the concepts "activation status of PBMC" and "CD4 + CD25 + subpopulations" must be explained.

INTRODUCTION

  • Lines 83-84: authors name the importance of IL-2: do they mean IL-2 secreted by MSC or exogenous IL-2 added in culture media?
  • lines 86-95: among the mentioned aspects, the authors should also consider the impact of "additives" for culture media such as FBS, Human Serum, Platelet Lysate etc...

MAT & MET

  • In the "materials and methods" paragraph the authors should specify the maximum passage in which ASCs are used for experiments.

RESULTS

  • In general the "Results" paragraph is difficult to understand, it must necessarily be revised and made more fluent.
  • Lines 232-234: PBMC monoculture is "transwell" with respect to what? CD3/CD28 stimulus?
  • Lines 234-236 are hardly understandable.
  • A curiosity/suggestion regarding stimulated PBMC – ASC co-cultures: have the authors plan to evaluate the effect of ASCs on PBMC after stimulus removal?
  • In the description of the results the authors mention "stimulated ASC co-cultures": are ASCs or PBMCs that have been stimulated?
  • Figure 1 (the authors must review the proposed graphs): what's the difference between A and A '? Explain. Figures 1B-1B' did not add value to what is proposed in A-A'. Simplify.
  • lines 345-347: this sentence is confusing and does not add values to the proposed results.
  • Lines 366-368 are hardly understandable.
  • Figure 3 B’-B’’: I suggest the authors to add statistical analysis between stim/ASC -/- (columns 1-2) and -/+ (column 5-6) both for proliferated and not proliferated CD4+CD25+. The trend does not seem so different between the two proposed graphs.
  • I suggest to the authors to add (in the supplementary materials section) some tables reporting the average percentages of positivity and SDs.
  • lines 445-448: "Treg fraction was lowered" respect to what?

DISCUSSION

  • lines 579-581: the authors postulate that ASCs induce CD127 on CD4+CD25+. What is the biological significance?
  • line 583: the authors support the hypothesis that the expression of CD127 on CD4+CD25+ is induced by IL-7 secreted by ASC. Was IL-7 production measured in ASCs alone?
  • Line 586: the authors argue that CD127 is closely dependent on the availability of IL-7 and IL-2. In this experimental setting did authors referred to exogenous IL-2 (added in the culture medium)?

Round 2

Reviewer 2 Report

The authors have provided all the answers to the objections raised. The article is suitable for publication